# A New Flattened Cylinder Specimen for Direct Tensile Test of Rock

**DOI:** 10.3390/s21124157

**Published:** 2021-06-17

**Authors:** Qiuhua Rao, Zelin Liu, Chunde Ma, Wei Yi, Weibin Xie

**Affiliations:** 1School of Civil Engineering, Central South University, Changsha 410075, China; raoqh@csu.edu.cn (Q.R.); zelinl@csu.edu.cn (Z.L.); yi.wei@csu.edu.cn (W.Y.); 2School of Source and Safety Engineering, Central South University, Changsha 410083, China; weibin@csu.edu.cn; 3Advanced Research Center, Central South University, Changsha 410083, China

**Keywords:** flattened cylinder specimen, clamp device, direct tensile test, tensile strength, rock

## Abstract

In recent decades, researchers have paid more attention to the indirect tensile test than to the direct tensile test (DTT) of rocks, mainly due to difficulties in the alignment and the stress concentration at the end of an intact cylindrical specimen. In this paper, a new flattened cylinder specimen and a clamp device were designed to obtain the true tensile strength of the rock in DTT. Stress distributions of the specimen with different lengths (*l*) and cutting thicknesses (*t*) were analyzed, and damage processes of the specimen were monitored by the Digital Image Correlation (DIC), the fractured sections were also scanned. Different mechanical parameters were also obtained by the DTT of the flattened cylinder specimens and the intact cylinder specimens, as well as the Brazilian disc. Research results show that the tensile strength obtained by DTT is smaller than the Brazilian disc and is slightly greater than the intact cylindrical specimen. The flattened cylinder specimen with 0.20 ≤ 2*t*/*D* < 0.68 and 0.10 ≤ *l*/*D* ≤ 0.20 is recommended to measure the true tensile strength of rock material in DTT. This new shape of the specimen is promising to be extended in the uniaxial or triaxial direct tension test.

## 1. Introduction

Tensile strength is one of rock’s important parameters, representing the ability of the rock to resist failure. Although natural rock is subjected to compressive loading in most cases, tensile stress still occurs during excavation unloading of tunnels, roadways, chambers, and slopes, etc. [1,2,3,4,5]. Since rock’s tensile strength is much smaller than its compressive strength [6,7], it plays a key role in controlling the stability of rock engineering, e.g., roof fall and rib spalling [8,9,10]. Therefore, studying the tensile strength of rock has great scientific and engineering significance.

Currently, there are mainly two testing methods for the tensile strength of rock: the indirect tensile test and the direct tensile test (DTT). The Brazilian test (BT) method was first proposed by Carneiro in 1943 and became an ISRM suggested method for indirectly determining the tensile strength of rock in 1978 [11,12]. Since then, a lot of research work has been done on the Brazilian test. Some solutions to both the stress and the displacement fields were obtained by the Brazilian circular disc based on the consideration of some factors like friction and thickness [13,14,15,16,17,18], and the tensile strength calculation formula (*σ_t_* = 2*p*/*πdt*) was modified by studying the influence of loading angle and Poisson’s ratio on the tensile strength of rock [19,20]. However, experimental observation of the failure process [21,22,23] showed that it is inevitable for the Brazilian disc specimen to fail to initiate at the loading point rather than at the center of the disc. In this case, the test results of rock tensile strength might deviate from their true value, and it is suggested that they be discarded [24]. Therefore, a flattened Brazilian disc with a loading arc of 20° was proposed to facilitate its failure, initiated at the center of the disc for indirectly measuring the tensile strength of rock [25,26]. Although the indirect tensile test is widely used, some researchers [27,28,29] have still raised doubts about whether it could be applied to measure the true tensile strength of rock, since the tensile strength determined by the indirect Brazilian test (BT) is usually larger than that by the direct tensile test (DTT).

Although ISRM [12] also suggested a direct tensile test (DTT) of a standard cylindrical specimen to determine the tensile strength of rock by a linkage system, keeping the axial alignment and eliminating the stress concentration at the end of the specimen became key problems for DTT. Otherwise, the specimen failure easily occurs at the end of the cylinder rather than in the middle area, which would result in a certain error between the tested and true tensile strength [29]. Therefore, many efforts were made on the loading fixture and specimen’s shape for DTT [30,31,32,33,34,35]. For example, Tufekci designed a new jaw clutch mechanism and dumbbell-shaped travertine specimen to determine the tensile strength of rock under uniaxial tension. Stimpson proposed a new testing technique in which moduli and strength in both tension and compression can be measured on the same specimen. Ramsey adopted a new dog bone-shaped marble specimen to measure the tensile strength of rock under uniaxial and triaxial tension. Due to difficulty in processing both the dumbbell-shaped specimen and dog bone-shaped specimen, it is necessary to seek a more suitable specimen for DTT.

This paper aims to get a simple processing rock specimen to measure the true uniaxial tensile strength in the direct tensile test, and a suggested method is proposed to measure rock tensile strength in a uniaxial or triaxial direct tensile test.

## 2. New Specimen and Clamp Device for DTT

### 2.1. Design and Preparation of a New Flattened Cylinder Specimen

A new flattened cylinder specimen is proposed for DTT, as shown in Figure 1. It was made by a standard cylinder (φ50 mm × 100 mm) cutting off two symmetric concaves (EACG and FBDH) symmetric to the arbitrary diameter symmetric plane O′O″. One concave (EACG or FBDH) was composed of two symmetrical arc surfaces EA and CG (with vertical distance *l_1_* from top and bottom ends of the specimen) for reducing the stress concentration at the ends of the specimen, and one flat surface AC (with length *l* and cutting thickness *t*) in the middle areas for obtaining a uniform stress distribution.

To prepare the flattened cylinder specimen (Figure 1) automatically by CNC machine, we (1) fixed the two ends of the cylindric specimen and determine the four points E, F, G, and H by *l*_1_; (2) determined the positions of the four points A, B, C, and D by *t* and *l* and the center O; and (3) cut off the arc surfaces EA/CG/FB/DH and flat surface AC/BD, meanwhile, smooth connection at the points A, B, C, and D to form the concave (EACG/FBDH), i.e., the flattened cylinder specimen.

To verify the validity of the new flattened cylinder specimen for DTT, FLAC3D software was adopted to calculate its maximum principal stresses (*σ*_1_) for different cutting thicknesses *t* (*t* = 0, 3 mm, 5 mm, 7 mm, 9 mm, 11 mm, 13 mm, 15 mm, 17 mm, 19 mm, 21 mm) and flat lengths *l* (*l =* 0, 5 mm, 10 mm, 15 mm, 20 mm, 30 mm, 40 mm, 50 mm, 60 mm) as shown in Figure 2, where the specimen model was meshed by a tetrahedral grid with a length of 2 mm and the tensile stress was selected as 10 MPa. For a given flat length *l* (*l* = 10 mm), as shown in Figure 2, when *t* = 0, *σ*_1_ is mainly concentrated on the end of the specimen and could easily cause the failure occurring on the specimen ends. When *t* = 3–5 mm, *σ*_1_ is concentrated in the platform area and is unevenly distributed. When *t* > 7 mm, *σ*_1_ is mainly distributed in the middle of the platform area. Considering the inconvenience of processing the specimen with *t* > 13 mm (*d*/*D* < 1/2), the suitable and suggested cutting thickness *t* is within 7–11 mm for the flattened cylinder specimen.

For a given cutting depth (*t* = 10 mm), as shown in Figure 2b, when *l* is quite small (*l* < 5 mm), *σ*_1_ is symmetrically distributed in the middle of the specimen but non-uniform. With the increase in *l*, *σ*_1_ gradually tends to be uniformly distributed in the middle area of the specimen. When *l* is much larger (*l* > 20 mm), *σ*_1_ is concentrated on both ends of the platform, which leads to negligible fluctuation in the failure position of the specimen. Therefore, the reasonable and suggested flat length *l* is within 10–20 mm for the flattened cylinder specimen in DTT.

### 2.2. Design and Preparation of Clamp Device

A special clamp device needs to be designed for DTT of the new flattened cylinder specimen by MTS815 testing machine (Figure 3a). It consists of upper and bottom contacting plates (Figure 3b), a tensile steel plate (Figure 3c), and a universal joint. The specimen is bonded to two bonding caps. The upper contacting plates and the universal joint transmit the force applied to the top end of the specimen to the force sensor located at the top of the specimen, while the universal joint can adjust the position of the specimen for alignment. In order to apply tensile stress to the lower end of the specimen by the actuator of the MTS518 testing machine, the bottom connecting plate, the loading plate, and the tensile steel plate are connected to the actuator by socket head cap screws and a right-angle fixing device to form a body, so that the specimen can be moved downward simultaneously with the actuator.

The triaxial chamber of the MTS815 test machine has its own alignment function to align the loading platform with the force sensor, and the bottom connecting plate is aligned with the loading platform through the designed alignment hole. In this way, we can ensure that the specimen is well aligned and without twisting during tests to the maximum extent possible. Furthermore, there is a seal ring on the contacting block, which could fit closely together with the heat-shrinkable tube to effectively separate the specimen from the oil in the triaxial tensile test.

## 3. Direct Tensile Test

### 3.1. Test Arrangement

In this study, fine-grain red sandstone extracted from the Yunnan Province of China was selected for DTT. It is approximately homogeneous and continuous, with a density of 2596 kg/m^3^. The flattened cylinder specimens of ϕ50 mm × 100 mm (Figure 1) were prepared with the same *l*_1_/*L (l*_1_/*L* = 1/10, i.e., *l*_1_ = 10 mm) and *l*/*L (l*/*L* = 1/5, i.e., *l* = 20 mm) and different *t* (*t =* 3 mm, 5 mm, 7 mm and 10 mm) and were named as A-number(cutting depth)–number(specimens’ order). All specimens were carefully polished for satisfying the precision requirement recommended by the ISRM [36]. In order to compare test results of the flattened cylinder specimens with those of other ISRM-suggested specimens, intact cylindrical specimens (ϕ50 mm × 100 mm) and Brazilian discs (ϕ50 mm × 25 mm) were also prepared for DTT and BT and were named as S-number (specimens’ order) and BT–number (specimens’ order), respectively.

All of the tests were conducted on the MTS815 servo-controlled testing system (with the maximum axial force of 2600 kN) under displacement control at a rate of 0.12 mm/min from the bottom of the specimens. Figure 4 depicts where the clamp device (Figure 4b) was used for the flattened cylinder specimens and the intact cylinder specimens, and the MTS tension fixture for the Brazilian discs (Figure 4c). In DTT, since the flattened cylinder specimens needed to be bonded to the upper and lower bonding blocks by the adhesive of 3 M Scotch-Weld before testing, the diameter of the bonding block needed to be slightly larger than the diameter of the specimen [12] and the bonding time is at least 12 h to ensure sufficient bonding strength (not less than 20 MPa). Stress–strain curves were automatically measured during the tests and failure patterns were recorded after the tests.

Digital Image Correlation (DIC) was also used to monitor the movement path of spots on the specimen’s surface to obtain the strain evolution for characterizing the entire failure process of DTT [37]. In this study, a charge-coupled device camera (Basler PiA2400-17gm) was installed perpendicularly to the specimen to capture the trail of spots. It has a resolution of 2448 × 2050 pixels with 8-bit digitization for grey levels, and a length–pixel ratio of the imaging system 0.0935 mm/pixel. In order to record the changes in the surface spot movement more clearly, the camera is set to take 15 pictures per second, and thus the strain field of the specimen can be calculated by the VIC-3D7 software system automatically.

### 3.2. Test Results and Discussions

#### 3.2.1. Stress–Strain Curve

Figure 5 shows the stress–strain curves of specimens in different tensile tests. For the flattened cylinder specimens (Figure 5a, except for the invalid specimens A-3-1, A-3-2, and A-3-3 due to breaking at the bonding surface of the adhesive and specimen) and the intact cylinder specimens (Figure 5b) in DTT, all the strain–stress curves (*t =* 5 mm, 7 mm, 10 mm) have similar characteristics. That is, they can be divided into three stages—linear elastic deformation stage (O-A), non-linear deformation stage (A-B), and failure stage (B-C)—without any compaction section. For the Brazil discs in BT (Figure 5c), the strain–stress curves are typically S-shaped and can be divided into three stages (O-A-B-C): the compaction stage (O-A), linear elastic stage (A-B), and the failure stage (B-C).

Table 1 lists test results of DTT and BT for different specimens, including Young’s modulus *E* and tensile strength *σ_t_*, as well as their average values. For the flattened cylinder specimens, as shown in Figure 6, the average *E* is decreased with the increase in *t,* while the average *σ_t_* is decreased first (*t* < 7 mm) and then slightly increased (*t* > 7 mm). The average *E* is much smaller than that of the intact cylinder specimens, since the cross-sectional area in the middle of the flattened specimen is smaller than the intact cylinder specimen at the same tensile stress; this will cause a larger deformation, resulting in a smaller *E*. The average *σ_t_* is slightly larger than that of the intact cylinder specimen but much smaller than that of the Brazilian disc specimen BT-1~BT-3. It might be that neglecting of the frictional stresses and intermediate principal stress (*σ*_2_) in the Brazilian test thus led to a larger *σ_t_*; the tensile strength is equal to 138.68%, 100.08%, and 118.52% of the tensile strength obtained by the intact cylinder specimen in DTT (2.43 MPa) and 69.48%, 50.51%, and 59.38% of the Brazilian test strength (4.85 MPa). This is consistent with Perras [29], which considered *DTS* = 0.7*BTS*, where *DTS* is direct tensile strength, *BTS* is Brazilian tensile strength, and 0.7 is the factor-dependent on sedimentary rocks.

#### 3.2.2. Failure Mode

Figure 5 shows the failure modes of specimens in different tensile tests. For the flattened cylinder specimens in DTT, the failure zones changed from the end area when *t* = 3 mm (Figure 5a) to the central platform area when *t* = 5 mm, 7 mm, and 10 mm. This means that larger cutting depths could effectively avoid the stress concentration at the end of the specimen and enable the tensile stress to be uniformly distributed on the central platform area, which is consistent with the numerical results (Figure 2). The flattened cylinder specimens are proven to be effective for the direct measurement of rock tensile strength.

For the intact cylinder specimens in DTT (Figure 5b), their failures occur near the loading end and the obtained rock strength may be slightly lower than the true strength due to stress concentration at the end. For the Brazilian disc specimens in BT (Figure 5c), their failures were along the diameter planes of maximum tensile stress (i.e., perpendicular to the loading direction) and could be valid for measuring rock tensile strength.

#### 3.2.3. Failure Mechanism

Figure 7 shows the strain evolution of the specimens in DTT monitored by digital image correlation (DIC) of the fractured sections from 20%–40%–60%–80%–100% of the peak stress(*σ_t_*) to the final failure, where the color (from purple to red) represents the value of strain fields (from small to large). More attention is paid to the strain evolution in the bottom of the specimen, since all tests are loaded by displacement control from the bottom of the specimens. For the specimen of *t* = 5 mm (Figure 7a), when the tensile stress is 20%*σ_t_*, the bottom arc surface zone (i.e., CG and DH in Figure 1) has an obvious large tensile strain and the middle platform zone (i.e., AC and BD in Figure 1) has a small concentrated tensile strain. When the tensile stress is 40%*σ_t_*, the large tensile strain field moves up to the bottom of the platform zone. As the tensile stress continues to increase (60%*σ_t_*–80%*σ_t_*–100%*σ_t_*), the large tensile strain field is gradually enlarged from the bottom arc surface zone to the platform zone and finally concentrated at the middle platform zone for tensile failure. The specimen of *t* = 7 mm (Figure 7b) has similar strain evolution to that of *t* = 5 mm. Regarding the specimen of *t* = 10 mm (Figure 7c), when the tensile stress is equal to 20%σ*_t_*, the specimen holds a large tensile strain in the bottom arc surfaces zone. With the increase in the tensile stress (40%*σ_t_*–60%*σ_t_*–80%*σ_t_*), the larger tensile strain field is gradually enlarged from the bottom arc surface zone to the platform zone and finally concentrated between the bottom platform zone and bottom arc surface zone. Therefore, tensile failure eventually occurs in the transition region of the specimen (near point B in Figure 1).

In addition, Figure 7 also shows the micro-topographies of the fractured sections in DTT, measured by ATOS TRIPLE SCAN Optical 3D scanner (Germany), and Table 2 lists their micro-characteristic parameters. It can be found that the maximum heights of the different specimens (with *t* = 5 mm, 7 mm, and 10 mm) slightly fluctuated from 3799 μm to 4229 μm on the whole. Since the ratio of scanning volume to scanning area (*V*/*S*) can represent the fluctuation of the fractured section, a small value of *V*/*S* (*V*/*S* = 1.18 mm, 1.24 mm, 1.12 mm) indicates the relatively uniform distribution of average heights of the fractured sections. Furthermore, the fractured section is approximately perpendicular to the direction of the tensile stress, i.e., typical tensile failure.

## 4. Conclusions

(1)A new flattened cylinder specimen with a curved-flat surface was successfully designed to measure the tensile strength of the rock in the direct tensile test (DTT). Compared with the current intact cylinder specimens in DTT, it has the advantages of reducing the end stress concentration, facilitating the uniform distribution of the maximum stress in the middle platform area of the specimen, and easily machining the specimen.(2)The self-designed clamp device for the new flattened cylinder specimen consists of the upper and bottom contacting plates, a universal joint, and a tensile steel plate. It can ensure that the flattened cylinder specimen is automatically centered to avoid eccentric stretching. It can be further extended for the triaxial tensile test.(3)For the same sandstone, the tensile strength measured by the flattened cylinder specimens in DTT is about 1.0–1.4 times and 0.5–0.7 times as large as that measured by the intact cylinder specimens in DTT and by the Brazilian disc specimens in BT, respectively, and is considered to be reasonable and effective. The new flattened cylinder specimen can be used to measure the tensile strength of rock for a direct tensile test.(4)The suggested sizes of the flattened cylinder specimens are:Diameter: *D* = 50 mmHeight to diameter ratio: *H*/*D* = 2.0–3.0.Vertical distance *l*_1_ to the height: *l*_1_/*H* ≥ 1:10.The flat length *l* to the diameter *D*: *l*/*D* = 0.1–0.2.The cutting thickness *t* to the diameter: *t*/*D* = 0.1–0.15.

## Figures and Tables

**Figure 1 sensors-21-04157-f001:**
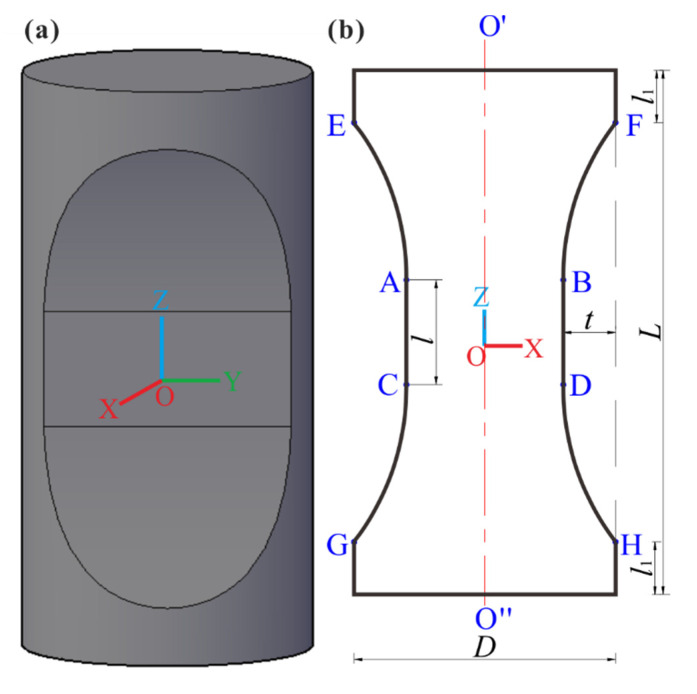
Schematic diagram of the new flattened cylinder specimen (**a**) stereogram (**b**) sectional view of *yoz* plane.

**Figure 2 sensors-21-04157-f002:**
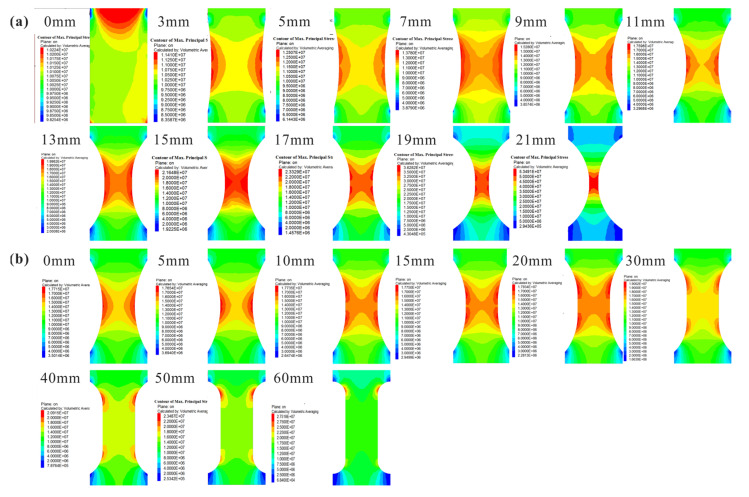
Maximum principal stress contours of the flattened cylinder specimens in *yoz* section plane (**a**) with different cutting thicknesses *t* (*t* = 0, 3, 5, 7, 9, 11, 13, 15, 17, 19, 21 mm) for a given flat length *l* (*l* = 10 mm) (**b**) with different flat lengths *l* (*l* = 0, 5, 10, 15, 20, 30, 40, 50, 60 mm) for a given cutting thickness *t* (*t* = 10 mm).

**Figure 3 sensors-21-04157-f003:**
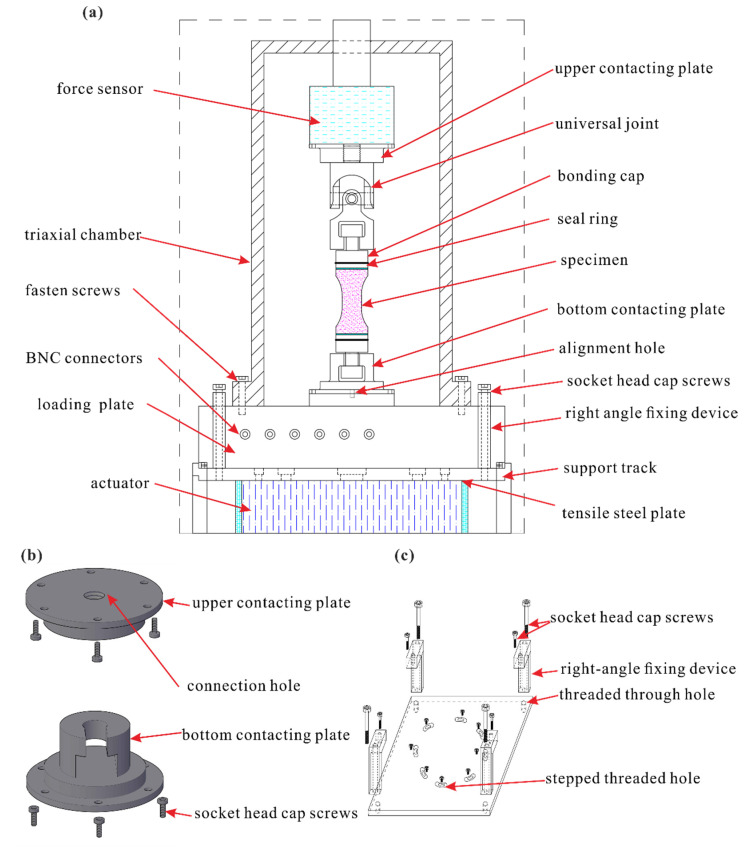
Test machine transformation: (**a**) MTS815 testing machine with a clamp device, (**b**) upper and bottom contacting devices, (**c**) tensile steel plane and its fixing device.

**Figure 4 sensors-21-04157-f004:**
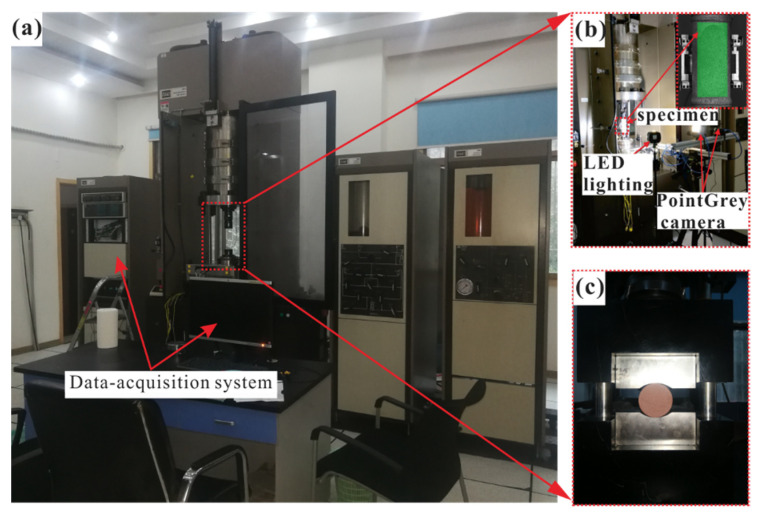
Test machine and loading setups: (**a**) MTS815 test system, (**b**) the direct tensile test with digital image correlation, (**c**) Brazilian test through MTS tension fixture.

**Figure 5 sensors-21-04157-f005:**
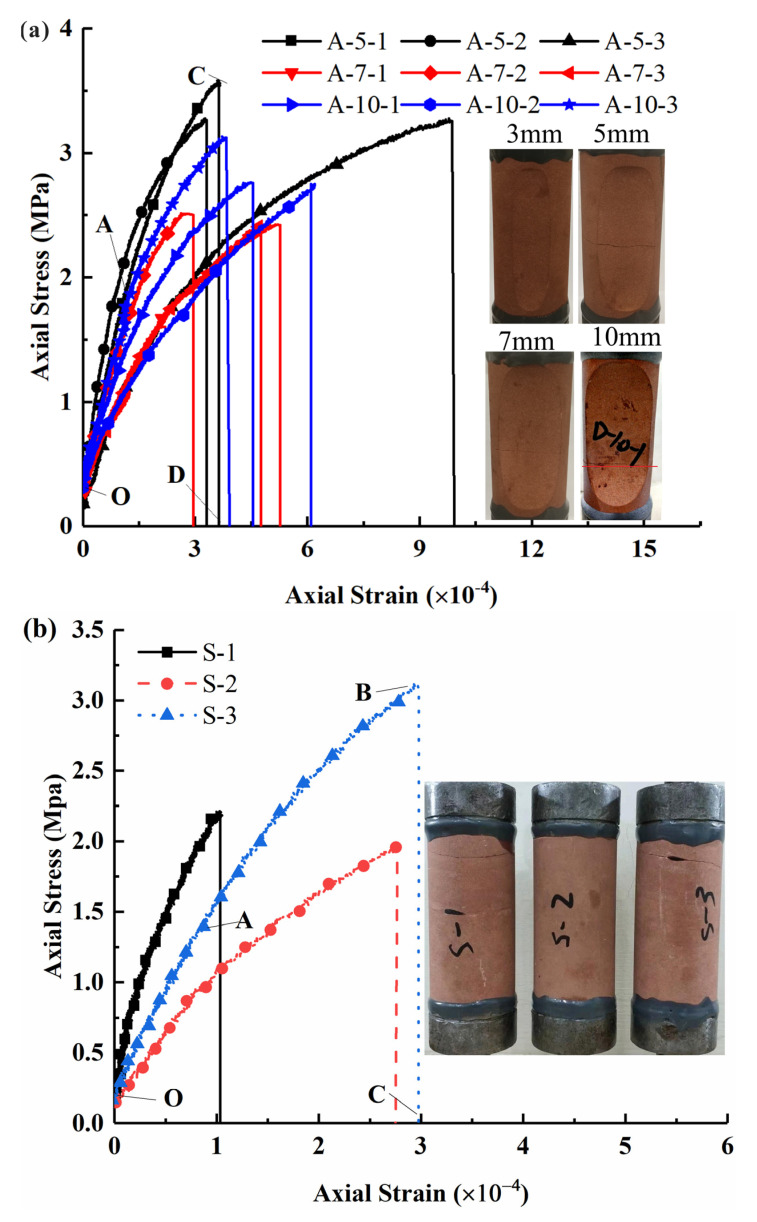
Stress–strain curves and failure modes of specimens in uniaxial tensile test, (**a**) DTT of flattened cylinder specimen, (**b**) DTT of intact cylinder specimens, (**c**) BT of Brazilian disc.

**Figure 6 sensors-21-04157-f006:**
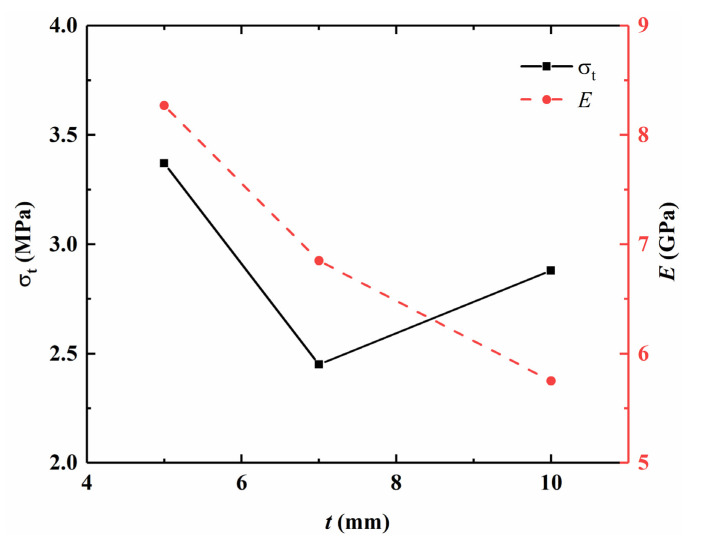
Variation of tensile stress and elasticity modulus with different cutting depths.

**Figure 7 sensors-21-04157-f007:**
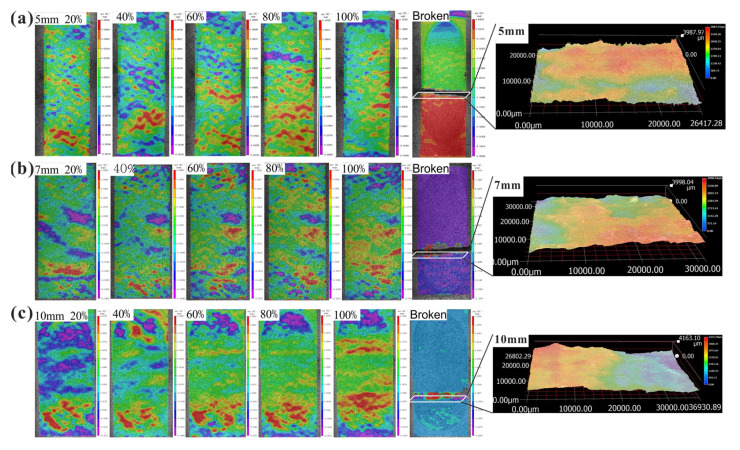
Strain evolution of the specimens in DTT monitored by digital image correlation (DIC) and micro-topographies of the fractured sections: (**a**) *t* = 5 mm, (**b**) *t* = 7 mm, (**c**) *t* = 10 mm.

**Table 1 sensors-21-04157-t001:** Test results of sandstone specimens in DTT and BT.

Test Type	No.	*D* (mm)	*H* (mm)	*σ**_t_* (MPa)	Mean (MPa)	*E* (Gpa)	Mean (GPa)	*ε*_max_ (×10^−4^)	Mean (×10^−4^)
DTT	S-1	49.78	100.28	2.21	2.43	22.14	13.58	1.03	2.24
	S-2	49.72	100.20	1.96		6.36		2.71	
	S-3	49.76	100.12	3.12		12.24		2.98	
	A-5-1	48.66	99.98	3.58	3.37	9.77	8.27	3.66	5.55
	A-5-2	48.70	100.12	3.28		9.89		3.31	
	A-5-3	48.68	100.06	3.26		5.15		9.69	
	A-7-1	48.54	100.02	2.42	2.45	5.98	6.85	5.19	4.30
	A-7-2	48.70	100.00	2.52		9.63		2.95	
	A-7-3	48.72	100.08	2.40		4.95		4.76	
	A-10-1	48.70	100.06	2.77	2.88	6.94	5.75	4.47	4.83
	A-10-2	48.70	99.98	2.75		4.40		6.20	
	A-10-3	48.68	100.12	3.13		5.91		3.83	
BT	BT-1	49.04	24.98	4.89	4.85				
	BT-2	49.10	24.98	4.90					
	BT-3	49.04	24.94	4.75					

**Table 2 sensors-21-04157-t002:** Micro-characteristic parameters of the fractured section from DTT.

Specimen	Cutting Depth (*t*) (mm)	Scanning Area (*S*) (mm^2^)	Maximum Height (*h*) (μm)	Scanning Volume (*V*) (mm3)	Scanning Volume/Scanning Area (mm)
A-5-1	5	528.34	3988	705	1.33
A-7-1	7	900.00	4229	1745	1.94
A-10-2	10	989.80	3799	2054	2.08

## Data Availability

Not applicable.

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
