# Peer review of "A New Flattened Cylinder Specimen for Direct Tensile Test of Rock"

_sensors, 2021, doi:10.3390/s21124157_

Round 1

Reviewer 1 Report

This article takes the real tensile strength of rock specimens as the starting point, and discusses the relationship between the direct and indirect tensile test results of the flattened cylinder specimens and the traditional specimens based on the new specimens form and the independently developed test equipment, which has a certain degree of innovation. At the same time, the selection of the new type of test piece is based on numerical simulation results first, and then mutual verification by experiments. The logic is relatively strict and the conclusions are reliable.

However, this paper has certain problems in the way of scientific expression and presentation, such as unclear test conditions, unclear diagrams of the paper, and exaggerated conclusions of the paper. It is recommended that the paper should be accepted after minor revision.

The main issues of the paper are as follows:

  1. The test conditions are not clear, whether it is a triaxial tensile test or a uniaxial tensile test is not mentioned in the article.
  2. It is recommended to add related graphs to illustrate the conclusions of the experiment, such as the conclusions in lines 181-183.
  3. There are problems in the narrative of the paper. For example, the specific meaning of the specimens’ number is not clearly stated in the text, and there is no definition of din Table 2.
  4. It is recommended to clearly express the fracture surface or fracture line of various specimens. The picture is too small and it is not recommended to mark the failure line directly on the specimens.
  5. The curve in Fig.5 is not clear enough, and the thickness of the curve and the size of the symbol in Fig.5(b) are unreasonable.
  6. The picture in the description of failure mode in 3.2.2 failure modeis not clear enough, it is recommended to reorganize it.
  7. There is a punctuation error in 3.2.2 failure mode, that is, the period before whichon line 210 should be a comma.
  8. There are some problems in this paper sentences. For example, the diameter of the bonding block must be slightly larger than the specimen one [9] and ……(line 152),what does specimen onemean?
  9. There is a suspicion of exaggerating the conclusion. How does the new specimens apply to the triaxial test in conclusion(3)?

Author Response

Response to Reviewer 1 Comments

Question 1: The test conditions are not clear, whether it is a triaxial tensile test or a uniaxial tensile test is not mentioned in the article.

Responses: This manuscript is for the uniaxial tensile test. We have added “the uniaxial tensile test” in line 200.

Question 2: It is recommended to add related graphs to illustrate the conclusions of the experiment, such as the conclusions in lines 181-183.

Responses: We have added figure 6 in lines 203-204 to illustrate the conclusions of the experiment in lines 181-183.

Question 3: There are problems in the narrative of the paper. For example, the specific meaning of the specimens’ number is not clearly stated in the text, and there is no definition of d in Table 2.

Responses: We have added “and is named as A-number(cutting depth)-number(specimens’ order).”to line 142-143, and “and were named as S-number(specimens’ order) and BT-number(specimens’ order),” to line 147-148. Besides, the column of d in Table 1 should correspond to the height (H) of the specimen (line202), we have corrected it in, and the d is the thickness of the specimens after being cut, which can be calculated by d=D-2t.

Question 4: It is recommended to clearly express the fracture surface or fracture line of various specimens. The picture is too small and it is not recommended to mark the failure line directly on the specimens.

Responses: We have redrawn Figure 5 to make it bigger and clearer, and removed the failure line on the specimens. However, there is still a line near the fracture surface of the sample in Figure 5(a), because the computer crashed and we lost the original image.

Question 5: The curve in Fig.5 is not clear enough, and the thickness of the curve and the size of the symbol in Fig.5(b) are unreasonable.

Responses: We have redrawn Figure 5, it is clearer than before. The thickness of the curve is consistent with others, it is only the slight fluctuations in the data during the tension test that make it unreasonable.

Question 6: The picture in the description of failure mode in 3.2.2 failure mode is not clear enough, it is recommended to reorganize it.

Responses: We have redrawn Figure 7, it is 600dpi now to meet the reading requirements.

Question 7: There is a punctuation error in 3.2.2 failure mode, that is, the period before whichon line 210 should be a comma.

Responses: We have corrected the period to a comma in line 211.

Question 8: There are some problems in this paper sentences. For example, the diameter of the bonding block must be slightly larger than the specimen one [9] and ……(line 152), what does specimen one mean?

Responses: We are sorry that the wrong writing enables it to be difficult to understand, it has been corrected to “the diameter of the bonding block must be slightly larger than that of the specimen [9]”.

Question 9: There is a suspicion of exaggerating the conclusion. How does the new specimens apply to the triaxial test in conclusion(3)?

Responses: The triaxial tensile test can be achieved in the following steps. Firstly, apply a hydrostatic pressure to the specimen, then, axially tensile until the specimen fractures and breaks (Zeng et al. Triaxial extension tests on sandstone using a simple auxiliary apparatus. Int J Rock Mech Min Sci. 2019;120:29-40). In our study, the new flattened cylinder specimen and clamp device can be used for uniaxial tensile test, it will turned to the triaxial tensile testafter adding the confining pressure on the specimen, and indeed, we have carried out research on triaxial tensile test.

Reviewer 2 Report

  1. The manuscript is interesting enough, although the topic is not very relevant. I believe that the relevance can be increased by expanding the boundaries of this manuscript. I believe that the literature review should be added with articles on concrete tests, for example:
  • Chernysheva, N.V.; Lesovik, V.S., Drebezgova, M. Yu. Composite Gypsum Binders with Silica-containing Additives. IOP Conference Series: Materials Science and Engineering. 2018. 327(3), 032015.  doi: 10.1088/1757-899X/327/3/032015
  • - Haridharan, M.K., Matheswaran, S., Murali, G., Abid, S.R., Fediuk, R., Mugahed Amran, Y.H., Abdelgader, H.S. Impact response of two-layered grouted aggregate fibrous concrete composite under falling mass impact. Construction and Building Materials. Volume 263, (2020), 120628
  • Elistratkin M.Yu., Kozhukhova M.I. Analysis of the factors of increasing the strength of the non-autoclave aerated concrete // Construction Materials and Products. 2018. Volume 1. Issue 1. P. 59 – 68

2. In Figure 2, it is difficult to understand small numbers
3. Conclusions do not correspond to the tasks set

Author Response

Response to Reviewer 2 Comments

Question 1: The manuscript is interesting enough, although the topic is not very relevant. I believe that the relevance can be increased by expanding the boundaries of this manuscript. I believe that the literature review should be added with articles on concrete tests, for example:

  • Chernysheva, N.V.; Lesovik, V.S., Drebezgova, M. Yu. Composite Gypsum Binders with Silica-containing Additives. IOP Conference Series: Materials Science and Engineering. 2018. 327(3), 032015. doi: 10.1088/1757-899X/327/3/032015
  • - Haridharan, M.K., Matheswaran, S., Murali, G., Abid, S.R., Fediuk, R., Mugahed Amran, Y.H., Abdelgader, H.S. Impact response of two-layered grouted aggregate fibrous concrete composite under falling mass impact. Construction and Building Materials. Volume 263, (2020), 120628
  • Elistratkin M.Yu., Kozhukhova M.I. Analysis of the factors of increasing the strength of the non-autoclave aerated concrete // Construction Materials and Products. 2018. Volume 1. Issue 1. P. 59 – 68

Responses: We have added the recommended refs to this manuscript.

Question 2: In Figure 2, it is difficult to understand small numbers

Responses: We have redrawn Figure 2, it is 600dpi and more clear now.

Question 3: Conclusions do not correspond to the tasks set

Responses: Although conclusion (2) is a summary of the clamp device, it is also useful and relevant to complete the uniaxial tensile test and even the triaxial tensile test. We would like to keep it as part of the conclusions.
